# SELF-ABLATING TRANSFORMERS
# MORE INTERPRETABILITY, LESS SPARSITY

**Jeremias Ferrao**[*]
University of Groningen
`j.l.ferrao@student.rug.nl`

**Luhan Mikaelson**[*]
Independent
`luhan.mikaelson@gmail.com`

**Keenan Pepper**[*]
AE Studio
`keenanpepper@gmail.com`

**Natalia Perez-Campanero Antolin**
Apart Research

## ABSTRACT

A growing intuition in machine learning suggests a link between sparsity and interpretability. We introduce a novel self-ablation mechanism to investigate this connection ante-hoc in the context of language transformers. Our approach dynamically enforces a k-winner-takes-all constraint, forcing the model to demonstrate selective activation across neuron and attention units. Unlike post-hoc methods that analyze already-trained models, our approach integrates interpretability directly into model training, promoting feature localization from inception. Training small models on the TinyStories dataset and employing interpretability tests, we find that self-ablation leads to more localized circuits, concentrated feature representations, and increased neuron specialization without compromising language modelling performance. Surprisingly, our method also decreased overall sparsity, indicating that self-ablation promotes specialization rather than widespread inactivity. This reveals a complex interplay between sparsity and interpretability, where decreased global sparsity can coexist with increased local specialization, leading to enhanced interpretability. To facilitate reproducibility, we make our code available at `https://github.com/keenanpepper/self-ablating-transformers`.

## 1 INTRODUCTION

As machine learning systems are entrusted with increasingly complex tasks, our ability to understand their decision-making processes lags behind their growing capabilities (OpenAI, 2024; Gemini Team, Google, 2024). This disparity is especially evident in Large Language Models (LLMs) based on the transformer architecture (Vaswani et al., 2017), whose dense and interconnected structures challenge even the most sophisticated interpretability tools (Foote et al., 2023b; Bills et al., 2023; Conmy et al., 2023; Huben et al., 2024; Gao et al., 2024; Zhang & Nanda, 2024; Miller et al., 2024).

Much of the current research in interpretability for LLMs focuses on developing post-hoc methods, attempting to explain the behaviour of already-trained models (Ribeiro et al., 2016; Conmy et al., 2023; Foote et al., 2023b; Bills et al., 2023; Huben et al., 2024). While valuable, these approaches often provide only an approximate or incomplete understanding of the underlying mechanisms (Rudin, 2019). A more fundamental yet less studied approach involves designing models to be inherently more interpretable, an ante-hoc approach, where transparency is woven into the architecture itself (Slavin et al., 2018; Tamkin et al., 2023; Cloud et al., 2024; Liu et al., 2024).

A prevailing intuition in the field is that sparser models are more interpretable (Zhang et al., 2017; Frankle et al., 2019; Conmy et al., 2023; Huben et al., 2024). This notion stems from the idea that fewer active components lead to simpler, more easily understood computations. While this connection has been explored through various post-hoc analysis techniques (Conmy et al., 2023; Huben et al., 2024), the intentional induction of sparsity during training as a means to enhance

---

[*]Equal Contribution

interpretability in models remains relatively unexplored (Slavin et al., 2018; Tamkin et al., 2023). This gap highlights an opportunity to investigate how inducing sparsity during training, specifically within the transformer architecture, can directly impact model interpretability.

In this work, we introduce a novel self-ablation mechanism to investigate the relationship between sparsity and interpretability in transformers. Our approach enforces a k-winner-takes-all (kWTA) constraint (Majani et al., 1988), encouraging selective activation across neuron and attention units during training via learned gating weights. Like training wheels, this mechanism is only active while learning; during inference, the model architecture is identical to a standard transformer, preserving its efficiency. This targeted approach allows us to study how structured sparsity impacts the model's internal representations and interpretability, paying the computational price only during training while reaping the benefits during inference.

We evaluate our approach by training Small Language Models (SLMs) on the TinyStories dataset (Eldan & Li, 2023). Our results demonstrate that the self-ablation mechanism leads to empirically improved interpretability, as measured by several tests (Conmy et al., 2023; Foote et al., 2023b; Bills et al., 2023; Huben et al., 2024), without significantly degrading model performance. Surprisingly, we also observe a decrease in overall sparsity, as measured by higher L1 activation norms. This suggests that promoting focused specialization through self-ablation can be more crucial for enhancing interpretability than maximizing sparsity.

Our work offers several key contributions to the field of interpretable AI. First, it introduces a novel self-ablation mechanism that provides a new avenue for building more interpretable-by-design models. Second, it reveals a complex interplay between sparsity and interpretability, challenging the common intuition that sparsity necessarily leads to more transparent models (Conmy et al., 2023; Huben et al., 2024) by demonstrating that decreased global sparsity can coexist with increased local specialization, leading to enhanced interpretability.

## 2 RELATED WORK

Our research is situated within a growing body of research on interpretable models by design and sparsity.

### 2.1 CODEBOOK FEATURES

A closely related work to ours is that of Tamkin et al. (2023), who introduce "Codebook Features" to enhance interpretability. Their approach adds a bottleneck layer, termed a codebook, after each component in a transformer. These codebooks replace the original component's activations before being added to the residual stream. Each codebook is composed of a large set of learned code vectors. During a forward pass, only the top-k code vectors most similar to the original activation are selected, inducing a high degree of sparsity, as the codebook's dimensionality is significantly larger than the original components. The authors demonstrate that the learned code vectors often exhibit human-interpretable functions, and that intervening on these codes can predictably alter model behaviour. However, their method substantially modifies the standard transformer architecture, leading to reduced inference speed and limiting compatibility with existing frameworks. Our work builds upon this concept of learned sparse representations but introduces a method that preserves the original transformer structure, enhancing interoperability and maintaining inference efficiency. Furthermore, while Tamkin et al. (2023) primarily rely on qualitative assessments of interpretability, we provide quantitative, measurable indicators to rigorously evaluate the impact of our self-ablation mechanism on model interpretability.

### 2.2 GRADIENT ROUTING

Similar to codebook features, Cloud et al. (2024) introduce "Gradient Routing" for mechanistic supervision of neural networks. This technique modifies backpropagation using data-dependent, weighted gradient masks, allowing specific data points to update designated parts of the network. The authors show that this method can create partitioned representations, enable robust unlearning through targeted ablation, and achieve scalable oversight by localizing modules responsible for different behaviours. Unlike codebook features, gradient routing maintains the standard transformer

architecture and inference efficiency. Our work draws inspiration from their approach to integrating specialization into the model. However, we apply kWTA directly to activations within each transformer layer during the forward pass, enforcing sparsity and inducing modularity within the core computational pathway of the model, a crucial distinction for achieving the degree of interpretability and control that we demonstrate.

## 2.3 K - WINNER TAKES ALL (KWTA)

Winner-takes-all (WTA) mechanisms, and their generalization to k-winner-takes-all (kWTA), have a long history of study in the context of neural networks (Lazzaro et al., 1988; Majani et al., 1988). These mechanisms, inspired by competitive interactions observed in biological neurons, enforce a significant form of sparsity where only a limited number of neurons are allowed to be active at any given time. Much of the prior research on kWTA in artificial neural networks has focused on its ability to improve efficiency. These efforts primarily focus on performance gains such as reduced memory footprints or increased throughput (Lazzaro et al., 1988; Majani et al., 1988; Hunter et al., 2022). In contrast, our work approaches kWTA primarily from an interpretability standpoint, leveraging its ability to localize information and induce modularity within the network. Additionally, instead of implementing kWTA by modifying the activation function, we introduce additional gating weights which allow us to extend kWTA naturally to attention units and also enable the model to selectively control its activations.

## 2.4 DROPOUT

While both our self-ablating mechanism (using kWTA) and dropout (Srivastava et al., 2014) involve deactivating parts of the network during training, they operate under fundamentally different principles and objectives. Dropout stochastically drops out neurons, primarily acting as a regularizer to encourage robustness by forcing redundancy and preventing over-reliance on specific units. In contrast, our self-ablation mechanism employs a controlled, deterministic process based on learned relevance. It leverages learned gating weights and the kWTA constraint to selectively activate only the most pertinent components for a given input. Thus, rather than promoting generalized robustness through redundancy, our approach explicitly enforces selective activation and functional specialization, aiming to create more localized and interpretable computational pathways.

## 3 METHODOLOGY

Building on prior approaches to sparsity and interpretability, we introduce a novel self-ablation mechanism that uniquely combines the interpretability benefits of selective activation with the practical advantages of the standard transformer architecture. This mechanism can be conceptualized as a set of auxiliary training supports, analogous to training wheels on a bicycle, that are active only during the learning phase. Crucially, during this learning phase, the standard transformer weights and the parameters of the self-ablation mechanism (the gating weights) are trained jointly end-to-end, allowing the mechanism to influence the representations learned by the base model. Our approach dynamically controls component activation within both Multi-Layer Perceptron (MLP) and attention layers using a kWTA constraint applied via gating weights. Specifically, within the attention mechanism, these gating weights are applied to the output of each attention head after the query, key, and value transformations. Importantly, because these auxiliary supports are deactivated during inference, the resulting model is structurally identical to a standard transformer. This allows us to maintain interoperability with existing frameworks, facilitating easier testing and deployment.

### 3.1 SELF-ABLATION MECHANISM

The core purpose of our self-ablation mechanism is to introduce dynamic binary masks during the forward pass, ensuring that only the most relevant components for a given input are active. These masks are applied to both neuron and attention units within the transformer. While a top-k selection would ideally achieve this behaviour, it poses a challenge due to its non-differentiable nature. We circumvent this problem by employing a straight-through estimator that utilizes a softmax function during gradient computation. We calculate a dynamic threshold as the midpoint between relevant activations that identifies a natural separation point between relevant and irrelevant components. The

temperature parameter controls how sharply this distinction is made, essentially determining how binary our selection becomes. Specifically, for a set of sorted activation values $x$, we first identify the $k$-th largest activation, denoted as $x_k$, and the $(k+1)$-th largest activation, denoted as $x_{k+1}$, where $k$ is the number of largest components we want to select and keep active. We then calculate a dynamic threshold, $\gamma$, as the midpoint between these two values. To control the sharpness of the selection, we compute a dynamic temperature term, $T$, based on the difference between the $k$-th and $(k+1)$-th largest activations. The final selection weights for the $n$ ablation neurons are then computed using a softmax function with the calculated temperature. These relationships are summarized in the following equations:

$$\gamma = \frac{x_k + x_{k+1}}{2} \tag{1}$$

$$T = x_k - x_{k+1} \tag{2}$$

$$w_i = \frac{e^{\frac{x_i - \gamma}{T}}}{\sum_{j=1}^{n} e^{\frac{x_j - \gamma}{T}}} \tag{3}$$

During the forward pass, we apply a hard selection based on the top-$k$ activations of the gating neurons, effectively creating a binary mask for the components that need to be ablated. Lower values of $k$ induce a larger bottleneck in the layer, forcing the model to further specialize its neurons, while higher values of $k$ allow each neuron to react more generally. However, during backpropagation, we use the continuous, softmax-derived weights in Equation 3 to compute gradients, employing the straight-through estimator technique. This allows us to train the model end-to-end despite the non-differentiable nature of the top-k operation.

To train the self-ablation mechanism itself, we utilize an ablated loss term in addition to the standard cross-entropy loss on the model's output. The total loss is the sum of the cross-entropy loss on the clean (unablated) model output and the cross-entropy loss computed on the ablated model's output, using the threshold and temperature calculated in Equations 1 and 2, respectively. This combined loss signal is then used to update all trainable parameters in the model via backpropagation (using the straight-through estimator for the ablation gates), including both the original transformer weights and the newly introduced gating weights. Thus, the base language model learns in conjunction with the self-ablation constraints from the beginning of training. To facilitate the computation of the clean loss, our model incorporates a dual residual stream where we store the states of both the clean (no ablations applied) and ablated residual streams. This structure allows us to obtain both the ablated and unablated outputs in a single pass through the model, which are crucial for calculating the activations of the MLP and attention units for the ablated stream.

## 3.2 Model Architecture

Our implementation extends the base GPT-Neo architecture (Black et al., 2021), utilizing the TinyStories-3M (Eldan & Li, 2023) configuration with 8 transformer blocks. Our self-ablation mechanism introduces additional parameters for the learnable gating weights, increasing the total parameter count to 3.6 million. These gating parameters are trained simultaneously with the base GPT-Neo parameters throughout the training process. Further information regarding the architecture and constant time complexity of self-ablation is provided in Appendices A and C.

To understand the effects of localized versus global information on ablation decisions, we designed two variants of this mechanism: a "local" approach and a "global" approach (Figure 1). In the local variant (Figure 1b), ablation occurs independently within each transformer block, allowing fine-grained, localized control over component activation at each processing stage based on the local context. This design was motivated by the desire to avoid a flow of information in the opposite direction, i.e., later layers influencing ablation decisions of earlier layers and enabling a look-ahead mechanism. This design was also motivated by the desire to reduce the computations required for training models with the ablation mechanism; an issue that becomes of utmost importance in the adoption of this technique for larger models. Conversely, the global approach (Figure 1a) utilizes an initial forward pass through the entire network to gather information before making ablation

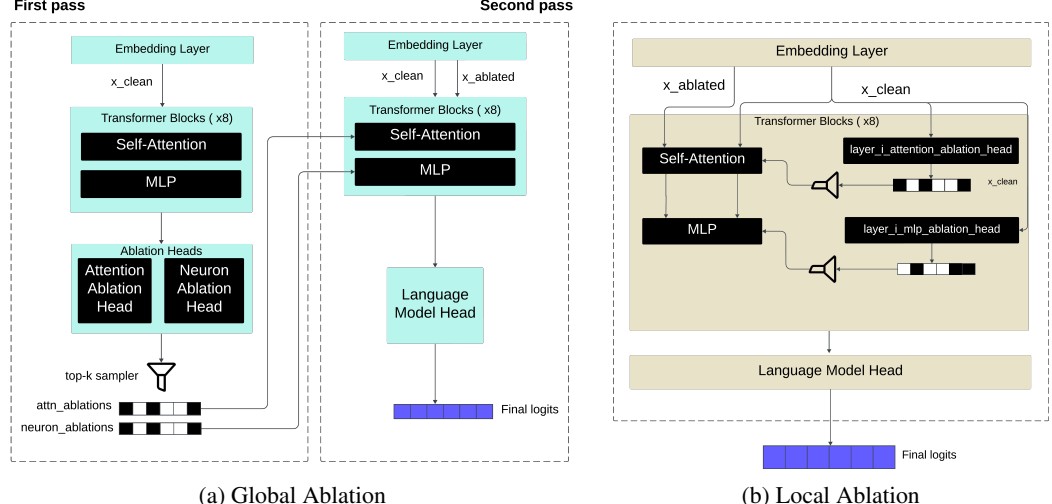

(a) Global Ablation         (b) Local Ablation

Figure 1: Comparison of global and local ablation mechanisms. Both models use a transformer with 8 blocks and are structurally identical during inference. Global ablation (a) uses a two-pass structure. The first pass calculates relevance scores using specialized ablation heads across the entire network. The second pass then processes the input using only the selected components based on these global scores. Local ablation (b) integrates the ablation mechanism directly into each transformer block. Each layer independently computes local relevance scores and makes ablation decisions based on its immediate context. See Appendix B for a detailed comparison of processing efficiency, information access patterns, and implementation considerations between the two approaches.

decisions for all layers simultaneously to be used for a subsequent pass. This global strategy is motivated by the inherent feature hierarchy in neural networks (LeCun et al., 2015), where complex representations emerge in later layers, potentially leading to more informed ablation based on a holistic view of the input.

## 3.3 TRAINING DATA

Our models are trained on the TinyStories dataset, a synthetic corpus of 2 million short stories (Eldan & Li, 2023). Each entry within the dataset was generated using a restricted vocabulary, mirroring the limited lexicon understood by 3- to 4-year-olds. This vocabulary constraint, coupled with simple grammatical structures, results in stories that are both semantically and syntactically uncomplicated. Although the simplicity of the dataset allows for rapid prototyping and training of language models, it also imposes limitations on the complexity of learned behaviours. We do not anticipate the emergence of highly intricate neural circuits within models trained on TinyStories. However, the dataset's inherent structure, which includes fundamental language concepts like subject-verb-object relationships, provides a suitable environment for the development of simpler circuits, such as Indirect Object Identification (IOI) observed in previous work (Wang et al., 2023; Conmy et al., 2023).

## 4 INTERPRETABILITY EXPERIMENTS

We employ a battery of tests designed to empirically assess the impact of our self-ablation mechanism on interpretability and sparsity. While these tests are not necessarily direct measures of the concept of interpretability, they nonetheless provide insights into the inner workings of our models. We compare our ablation-trained models against a baseline TinyStories 3M model without self-ablation.

### 4.1 AUTOMATIC CIRCUIT DISCOVERY

We utilize Automatic Circuit Discovery (ACDC) (Conmy et al., 2023) to identify key components within our models responsible for specific behaviours, providing a more localized understanding

of their internal computations. ACDC helps pinpoint crucial parts of the model for a given task, effectively creating a simplified sub-network or "circuit". In our work, we focus on the IOI task, where the model must identify the indirect object in a sentence. A crucial parameter in ACDC, $\tau$, determines the threshold for retaining a component in the circuit; a lower $\tau$ results in a sparser circuit. We set $\tau$ across all experiments to 0.03 for visualization purposes and to ensure a standardized evaluation procedure. As described by the authors, a circuit with fewer edges is considered more desirable because it is less likely to include irrelevant information and suggests a better localization of the concepts learned by the model. Further information regarding our ACDC experiments can be found in Appendix D.

## 4.2 Sparse Autoencoders

To investigate how self-ablation affects high-level concept representation, we employ Sparse Autoencoders (SAEs) to learn a simplified representation of the model's internal activations. We focus on the activations from the penultimate transformer block after the MLP layer. We trained these SAEs with an expansion factor of 16 until their total loss converged. We use the L0 norm as our primary sparsity metric, which directly measures the average number of non-zero feature activations in the SAE's bottleneck. A lower L0 score indicates each activation is being represented by a more selective set of features. To evaluate the SAE's ability for reconstruction, we adopt the cross-entropy loss score. This metric quantifies the change in model performance when the original activations are replaced with their SAE reconstructions during a forward pass. The cross-entropy loss score ranges from 0 to 1, with higher values indicating that the SAE's reconstruction preserves more of the original model's behaviour. These widely used metrics (SAE Bench, 2025), allow us to examine whether self-ablation leads to more disentangled and interpretable internal representations. We report the exact hyper-parameters used in our SAE experiments in Appendix E

## 4.3 Neuron Explainability

To assess neuron interpretability, we employ an automated approach by Bills et al. (2023), which utilizes an examiner LLM (in our case, GPT-4o mini (OpenAI, 2024)) to generate and score natural language explanations of neuron behaviour. While this automated process is still computationally expensive, it allows us to analyze neuron behaviour at a larger scale than manual methods. The process involves generating a natural language explanation of a neuron's behaviour using the examiner LLM, then using this explanation to simulate a hypothetical neuron's behaviour on new inputs. The correlation between the simulated and actual neuron activations serves as the "explanation score", quantifying the explanation's accuracy. Due to monetary and time constraints, we focused on analyzing the neurons of one of our self-ablated models: a global ablation model with $k$=2. We obtain a distribution of explanation scores by applying this process to every neuron within the model. A higher average explanation score across all neurons suggests better overall neuron interpretability, as it indicates more consistent and explainable behaviour.

## 4.4 Neuron to Graph

We use the Neuron to Graph (N2G) framework developed by Foote et al. (2023a) to analyze how self-ablation affects neuron behaviour in our ablated language models. Our methodology examines neuron behaviour through comparative analysis techniques.

For our aggregated analysis of the base and ablated models' graphs, we examine several metrics that help us understand neuron behaviour: graph density, out-degree, and graph transitivity. Graph density is calculated as the ratio of actual edges in the graph to the maximum possible number of edges; lower density indicates more selective neuron behaviour, as the neuron forms fewer connections between tokens. Out-degree is the average number of outgoing connections per node; higher values in later layers combined with low density can suggest better feature abstraction, maintaining meaningful connections while eliminating spurious ones. Graph transitivity measures the local clustering of related concepts/features; lower values indicate more selective behaviour with fewer interconnected activation patterns. Our analysis also focuses on activating nodes (tokens that trigger high neuron activation) to provide further insight into neuron specialization.

Table 1: Impact of Self-Ablation on interpretability and performance. Lower $k$-values in the self-ablation mechanism generally improve interpretability, despite a decrease in sparsity as indicated by higher L1 Norms. This suggests that while our models are less sparse overall, they still benefit from increased specialization, leading to improved interpretability. There is a minimal increase in perplexity compared to the regular transformer baseline. Arrows indicate the optimization direction: $\downarrow$ denotes that lower values are better for the metric.

| Architecture | | ACDC | SAE | | LM | |
|---|---|---|---|---|---|---|
| Ablation Type | K | IOI Edges ↓ | L0 Norm ↓ | CE Score ↑ | Perplexity ↓ | Sparsity (L1) |
| Baseline | - | 79 | 7.22 | 0.63 | **5.73** | 0.44 |
| Global | 8 | 70 | 5.22 | 0.65 | 6.97 | 0.74 |
| Local | 8 | 36 | 5.0 | **0.72** | 6.47 | 0.67 |
| Global | 4 | 41 | 4.9 | 0.67 | 6.73 | 0.72 |
| Local | 4 | **30** | **4.01** | 0.65 | 6.58 | 0.76 |
| Global | 2 | **30** | 4.57 | 0.71 | 6.66 | 0.73 |
| Local | 2 | 54 | 4.9 | 0.7 | 6.55 | 0.71 |
| Global | 1 | 38 | 5.48 | 0.66 | 6.49 | 0.62 |
| Local | 1 | 36 | 5.33 | 0.69 | 6.58 | 0.7 |

For token substitution analysis, we develop specialized analyzers that incorporate grammatical patterns for different token types. We use DistilBERT (Sanh et al., 2019) to generate contextually appropriate substitutions and test each suggested substitution to maintain similar activation patterns. We also examine the contextual environment of activating tokens to preserve grammatical structure. This analysis reveals how ablation affects neurons' responses to different token patterns. We perform a comparative analysis of substitution patterns across both models based on token-type diversity by layer and neuron specialization using entropy, allowing us to better understand how the ablation mechanism affects neuron interpretability.

## 4.5 LANGUAGE MODELLING

To evaluate the impact of our self-ablation mechanism on language modelling performance, we report the validation perplexity of our models. Perplexity, intuitively, measures how "surprised" a model is when it sees new text. A model with lower perplexity is better at predicting the next word in a sequence, indicating a better understanding of the language. This metric helps us understand whether the interpretability benefits of self-ablation come at a significant cost to the model's core language modelling capabilities. Furthermore, to determine the interplay between sparsity and interpretability, we measure the L1 norm, calculated as the normalized sum of the model's weights. Sparser models have a lower L1 norm, serving as a useful indicator of sparsity.

## 5 RESULTS

Our experiments demonstrate that self-ablation significantly enhances model interpretability across multiple evaluation methods, while incurring only a modest cost in language modelling performance. Synthesizing these findings reveals a consistent picture: self-ablation produces more localized functional circuits (evidenced by substantially fewer edges identified by ACDC (Conmy et al., 2023)), more concise and disentangled feature representations (indicated by lower L0 norms and strong reconstruction scores using SAEs (Huben et al., 2024; SAE Bench, 2025)), more readily explainable individual neuron behaviors (supported by higher automated explanation scores (Bills et al., 2023)), and more focused neuron activation patterns (reflected in sparser connectivity graphs and increased specialization metrics from N2G analysis (Foote et al., 2023a)). These improvements are observed compared to the baseline transformer, with quantitative details provided in Table 1 and Table 2. Importantly, this enhanced interpretability comes at a manageable cost to the model's core capabilities: validation perplexity increases by no more than 20% across all self-ablation configurations com-

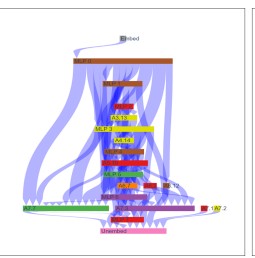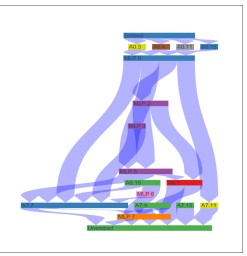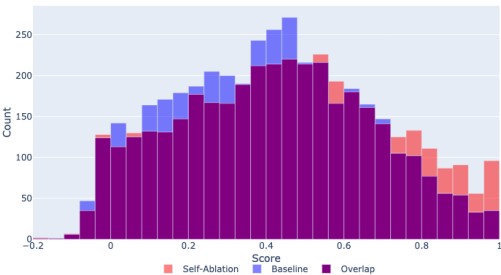

Figure 2: Self-ablation improves interpretability as shown through (left) IOI circuit simplification (baseline: 79 edges, self-ablated: 30 edges) where fewer model components are required to perform a task and (right) a shift towards higher neuron explanation scores in the self-ablated model (red) compared to the baseline (blue). These results indicate that self-ablation leads to more focused and interpretable circuits and neurons.

pared to the baseline (Table 1). The subsequent sections will elaborate on the specific findings from our circuit, feature, and neuron-level analyses.

## 5.1 CIRCUIT ANALYSIS

The effectiveness of our approach is most striking in the ACDC and SAE evaluations. The ACDC analysis shows that self-ablation dramatically reduces circuit complexity as observed in Figure 2. The most notable example is our local ablation model with $k$=4, which achieves an IOI circuit edge count of just 30, compared to 79 in the baseline - a 62% reduction. This substantial decrease in edge count indicates that self-ablation produces more localized and focused circuits. Such circuit simplification is particularly valuable for interpretability, as it makes it easier to trace and understand the specific computations underlying model behaviour.

## 5.2 FEATURE REPRESENTATION ANALYSIS

The SAE results provide further evidence for the interpretability gains achieved through self-ablation. Two key metrics highlight this improvement: the L0 norm and the Cross-Entropy (CE) score. The L0 norm, which measures the average number of active features in the SAE's bottleneck layer, is lower in our ablated models. For example, the $k = 4$ local ablation model achieves an L0 norm of 4.01, compared to the baseline's 7.22. This 44% reduction in active features indicates that self-ablation leads to sparser, more focused representations. Each feature captures a more specific aspect of the model's behaviour, suggesting a greater degree of disentanglement. Furthermore, our models consistently achieve higher CE scores, which quantify how well the SAE reconstructions preserve the original model's behaviour. The $k = 8$ local model attains a CE score of 0.72, surpassing the baseline's score of 0.63. This improvement demonstrates that despite using fewer active features, the ablated model's representations remain faithful to the original computation.

## 5.3 NEURON EXPLAINABILITY ANALYSIS

Our automated neuron interpretability analysis, focusing on a self-ablated model trained with global ablation and $k = 2$, reveals a shift towards higher explanation scores compared to the baseline model (see Figure 2). Quantitatively, the self-ablated model achieves a mean explanation score of 0.46, while the baseline model has a mean score of 0.41. This difference suggests that the neurons in the self-ablated model are, on average, more amenable to meaningful interpretation, supporting the notion that our self-ablation mechanism enhances neuron interpretability.

## 5.4 NEURON GRAPH ANALYSIS

Our NeuronGraph analysis reveals significantly more focused neuron behaviour in the ablated model compared to the baseline (Table 2). The baseline model exhibits consistently higher graph density

across all layers (0.235-0.446), while the ablated model shows significantly lower densities (0.046-0.191), indicating sparser, more focused connectivity patterns. This is further supported by the increase in activation node count, which measures connections leading to strongly activating tokens. In the baseline model, this metric remains high in later layers (1.30-1.57), while the ablated model shows a decreasing trend from 0.45 in Layer 0 to 0.07 in Layer 7, suggesting increasing neuron selectivity. Additionally, neuron specialization analysis based on the entropy of token substitutions demonstrates increased specialization in higher layers of the ablated model. The baseline maintains relatively high entropy in later layers (2.45-2.65), but the ablated model exhibits significantly lower values (1.14-2.16). This is corroborated by the token diversity analysis, where neurons in higher layers of the ablated model respond to fewer token types (3-5) compared to the baseline (7-8). These metrics collectively demonstrate that neurons in the ablated model develop more focused, specialized roles, responding to fewer types of tokens more consistently.

The relationship between transitivity, out-degree, and graph density reveals sophisticated changes in how neurons process information in the ablated model. The ablated model's lower transitivity, particularly in layers 2-7 where it approaches zero, indicates fewer interconnected activation patterns and more selective behaviour. Notably, the combination of higher out-degree in later layers (0.98-1.01) with significantly reduced graph density and edges to activation suggests that the ablated model maintains diverse, meaningful connections while eliminating spurious ones. This contrasts with the baseline model's stable but less selective connectivity, characterized by consistent out-degree (0.86-0.91) alongside higher density and more activating edges. This combination of metrics reveals that the ablated model achieves a crucial balance: it maintains meaningful connections (high out-degree) while eliminating spurious ones (low density and transitivity). This suggests that the model learns to detect specific, abstract features while filtering out noise - a key characteristic of interpretable representations.

## 5.5 LANGUAGE MODELLING PERFORMANCE

While self-ablation does lead to a slight increase in validation perplexity, indicating a trade-off between interpretability and raw language modelling performance, this increase is relatively modest. The largest increase is seen with global ablation with $k = 8$, where the perplexity rises from 5.73 (baseline) to 6.97. The local ablation models, however, maintain perplexity levels that are closer to the baseline, with the $k = 4$ local model achieving a perplexity of 6.58. Interestingly, despite the improvements in interpretability metrics like ACDC circuit size and SAE L0 norm, our self-ablation mechanism does not lead to sparser representations overall, as evidenced by the significantly higher L1 norm across all ablated models. This unexpected finding suggests that the increased interpretability may arise through mechanisms other than simply reducing overall sparsity, such as increased specialization and the development of more localized circuits. The improved interpretability, therefore, comes at a manageable cost to the model's predictive capabilities, but potentially through a different pathway than initially hypothesized.

## 6 DISCUSSION

This work introduced a novel self-ablation mechanism with global and local implementations designed to enhance transformer interpretability by enforcing a kWTA constraint during training. Our results demonstrate that self-ablation improves interpretability across various indicators, including ACDC circuit analysis, SAE feature analysis, and neuron graph analysis, albeit with a modest trade-off in perplexity. The local approach generally outperformed global ablation, likely due to its granular control at each network level. Local decisions also avoid potential information bottlenecks associated with global ablation, and the direct feedback inherent in local ablation training may promote more efficient learning. A key feature of both approaches is their role as auxiliary training support: the specialized ablation weights, active only during training, guide the model toward more interpretable representations, then are deactivated during inference, rendering the model structurally identical to a standard transformer and ensuring compatibility with existing frameworks.

Interestingly, while interpretability improved, our ablation-trained models exhibited lower overall weight sparsity (higher L1 norm). This finding runs counter to both a common intuition in the field and specific literature suggesting sparser representations are inherently more interpretable (Conmy et al., 2023; Huben et al., 2024; Slavin et al., 2018), often because fewer active components sim-

Table 2: Graph Structure Analysis Across Layers. The ablated model generally shows reduced connectivity and complexity metrics across all layers, indicating more focused and interpretable representations. Abbreviations: G.Den=Graph Density, Tran=Transitivity, E/A=Edges to Activation, O.D=Out Degree, N.Ent=Neuron Entropy, T.Div=Token Diversity.

| Metric | Model | L0 | L1 | L2 | L3 | L4 | L5 | L6 | L7 |
|---|---|---|---|---|---|---|---|---|---|
| G.Den ↓ | Base | 0.235 | 0.409 | 0.407 | 0.443 | 0.446 | 0.420 | 0.400 | 0.395 |
| | Ablated | **0.191** | **0.057** | **0.081** | **0.070** | **0.095** | **0.046** | **0.056** | **0.083** |
| Tran ↓ | Base | 0.000 | 0.001 | 0.001 | 0.001 | 0.001 | 0.001 | 0.003 | 0.006 |
| | Ablated | 0.000 | 0.001 | **0.000** | 0.001 | **0.000** | **0.000** | **0.000** | **0.001** |
| E/A ↓ | Base | 0.71 | 1.45 | 1.57 | 1.56 | 1.40 | 1.38 | 1.30 | 1.32 |
| | Ablated | **0.45** | **0.23** | **0.17** | **0.10** | **0.08** | **0.06** | **0.04** | **0.07** |
| O.Deg | Base | 0.88 | 0.91 | 0.86 | 0.86 | 0.86 | 0.86 | 0.86 | 0.89 |
| | Ablated | 0.78 | 0.74 | 0.92 | 0.98 | 0.98 | 0.98 | 1.01 | 1.00 |
| N.Ent ↓ | Base | 2.08 | 2.45 | 2.57 | 2.10 | 2.59 | 2.53 | 2.65 | 2.45 |
| | Ablated | 2.38 | 2.48 | 2.48 | 2.44 | **1.14** | **2.40** | **2.16** | **2.10** |
| T.Div ↓ | Base | 8 | 8 | 7 | 6 | 8 | 8 | 7 | 8 |
| | Ablated | 8 | 8 | 7 | 7 | **3** | **6** | **5** | **5** |

plify analysis. Our results, however, indicate that the interpretability gains stem from increased functional specialization and circuit localization—akin to local connectivity in CNNs (Lecun et al., 1998)—rather than merely reduced overall activity. Self-ablation thus appears to achieve interpretability through a mechanism focused on structured, localized computation, potentially distinct from traditional weight sparsity, highlighting a complex relationship worthy of further study.

## 6.1 LIMITATIONS AND FUTURE WORK

We acknowledge important limitations impacting the generalizability of our current findings. Our primary evaluation platform was the TinyStories dataset. While useful for rapid prototyping, its inherent linguistic simplicity—lacking complex syntax, diverse vocabulary, abstract reasoning, and significant ambiguity—means our models likely formed simpler circuits than required for real-world language (Gokaslan et al., 2019; Gao et al., 2020). Consequently, the observed interpretability benefits may not directly translate when models must handle these more complex linguistic phenomena. Furthermore, our experiments were conducted on relatively small models. It is currently unknown how the self-ablation mechanism, particularly its effect on circuit localization and feature specialization, scales to much larger architectures.

Addressing these limitations defines key directions for future research. Firstly, evaluating self-ablation on linguistically diverse datasets is essential to test scalability. Beyond standard benchmarks (Glazer et al., 2024; Wang et al., 2024), this requires targeted analysis to confirm whether interpretable representations are learned for the complex linguistic features and reasoning capabilities absent in TinyStories. Investigating the impact of model scale must also assess potential trade-offs, such as whether enforcing local selectivity hinders global coherence over long contexts, particularly when comparing local versus global ablation strategies.

Furthermore, the localized representations fostered by self-ablation suggest potential advantages in AI safety contexts. For instance, investigating how these models perform in controlled unlearning scenarios would be valuable, building on work like Cloud et al. (2024) and aligning with safety goals measured in benchmarks like WMDP (Li et al., 2024). Finally, given the significant cost of pre-training, exploring the application of self-ablation within a fine-tuning paradigm offers a practical avenue for improving the interpretability of existing large language models.

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

## A  MODEL TRAINING

This section details the hyperparameters used to train the models presented in this study. We employed the GPT-Neo architecture (Black et al., 2021) with the TinyStories-3M configuration (Eldan & Li, 2023) as our base model. The following table outlines the specific settings used during the training process. Due to variations in hardware and conditions used across different training runs, we do not report specific training times. Additionally, we have not optimized our training pipeline, which would considerably improve the feasibility of self-ablation training.

## B  LOCAL VS. GLOBAL ABLATION

We summarize the differences between our global and local self-ablation architectures in Table 4.

## C  SELF ABLATION TIME COMPLEXITY

Self-ablation dynamically controls the activation of components during the forward pass, effectively inducing a form of structured sparsity. While the mechanism introduces a sorting step, its time complexity remains comparable to that of a standard Multi-Layer Perceptron (MLP). The sorting operation has a complexity of O(N log N), where N is the number of units (neurons or attention

Table 3: Hyperparameters used for model training.

| Hyperparameter | Value |
|---|---|
| Model Architecture | GPT-Neo |
| Configuration | TinyStories-3M |
| Residual Stream Width | 128 |
| Number of Blocks | 8 |
| Number of Attention Heads | 16 |
| Attention Type | Global |
| Learning Rate | 0.0014 |
| Max Position Embedding | 256 |
| Optimizer | AdamW |
| Weight Decay | 0 |
| Learning Rate Scheduler | Cosine Annealing |
| Maximum Gradient Norm | 1 |
| Batch Size | 24 |
| Number of Training Iterations | 400,000 |

Table 4: Detailed comparison of global and local ablation mechanisms. This table outlines the key architectural differences and their implications for processing efficiency and implementation.

| Aspect | Global Ablation | Local Ablation |
|---|---|---|
| Processing Structure | Two-pass approach: initial pass for scoring, second pass for processing | Single-pass approach with integrated ablation |
| Computational Overhead | Higher - double forward pass requirement | Lower - single integrated pass |
| Information Access | Full network visibility; access to higher-level features | Limited to the current layer's context and previous layers' outputs |
| Decision Granularity | Network-wide decisions made simultaneously | Layer-specific decisions made sequentially |
| Feature Hierarchy Integration | Explicitly leverages complete feature hierarchy | Implicit through sequential processing |
| Implementation Complexity | More complex: requires coordination between passes | Simpler: self-contained within each transformer block |
| Scalability | May face challenges with very large models due to memory requirements | More scalable due to local computation pattern |

heads) being ablated. Since N is a constant determined by the model's architecture and not the input size, it does not significantly impact the overall time complexity for sufficiently large inputs. The primary computational cost remains dominated by the matrix multiplications inherent in the MLP and attention mechanisms, similar to the standard transformer architecture.

## D  AUTOMATIC CIRCUIT DISCOVERY

This section details our application of the Automatic Circuit Discovery (ACDC) algorithm, introduced by Conmy et al. (2023), to identify the Indirect Object Identification (IOI) circuit within our self-ablated models. We utilized the Auto Circuit library by Miller et al. (2024), to efficiently implement ACDC. To generate the IOI dataset, we adapted the generator script from the Auto Circuit repository. This script employs a set of predefined templates (e.g., 'BABA_TEMPLATES',

'ABBA_TEMPLATES') and a vocabulary of names, places, and objects, which we modified to align with the TinyStories dataset used to train our models. Prompts follow structures like "Then, [B] and [A] went to the [PLACE]. [B] gave a [OBJECT] to [A]", where '[A]' and '[B]' represent names, '[PLACE]' a location, and '[OBJECT]' an object. The script also generates corrupted versions of these prompts by swapping or replacing names.

We used the Kullback-Leibler (KL) divergence as the primary metric to evaluate the impact of activation patching during the ACDC procedure. This metric quantifies the difference between the model's output distribution on a clean prompt and its distribution when activations are patched from a corrupted prompt. A threshold parameter $\tau$ of 0.03 was used to control the sparsity of the discovered circuit, with lower values leading to circuits with more edges.

## E  SPARSE AUTOENCODER HYPERPARAMETERS

This section outlines the hyperparameters used for training our Sparse Autoencoders (SAEs). We employed the SAE lens library for training (Joseph Bloom & Chanin, 2024). The specific settings used in our experiments are detailed below. Note that we used a custom model, and any hyperparameters not mentioned here were set to the default values of the library.

Table 5: Hyperparameters used for SAE training.

| Hyperparameter | Value |
| --- | --- |
| Expansion Factor | 16 |
| $b\_dec$ Initialization Method | zeros |
| Apply $b\_dec$ to Input | False |
| Initialize Encoder as Decoder Transpose | True |
| Normalize Activations | expected_average_only_in |
| Learning Rate (lr) | $1 \times 10^{-5}$ |
| Learning Rate Scheduler | Constant |
| Learning Rate Warm-up Steps | 0 |
| Learning Rate Decay Steps | 20,000 |
| L1 Coefficient | 5 |
| L1 Warm-up Steps | 5,000 |
| Lp Norm | 1.0 |
| Training Batch Size (Tokens) | 4,096 |
| Hook Name | blocks.6.hook_mlp_out |
| Number of Batches in Buffer | 64 |
| Total Training Tokens | 409,600,000 |
| Store Batch Size (Prompts) | 16 |
| Seed | 42 |

