# OpenReview forum: "Self-Ablating Transformers: More Interpretability, Less Sparsity"
_ICLR.cc/2025/Workshop/BuildingTrust — BuildingTrust_

### Official Review · Reviewer_3sRk · 2025-03-02
**This paper introduces a self-ablation training mechanism for transformer using k-winner-takes-all during training—to show that focused specialization, rather than global sparsity, can enhance interpretability at a modest cost in perplexity and other core metrics.**

**Rating:** 8
**Confidence:** 4

**Review:**

This paper designs and evaluate a training mechanism that enforces selective activation in Transformer models. The authors introduce a self-ablation procedure based on a $k$-winner-takes-all method that only remains active during training, so that the final inference-time architecture is unchanged. The text is mostly clear, although the descriptions of local vs. global ablation could use a bit more elaboration. Nonetheless, the methodology is laid out in enough detail for another researcher to attempt a replication.

The paper’s main strength lies in its careful demonstration that pushing for “focused specialization,” rather than overall massive sparsity, can lead to more interpretable internal circuits. This is a useful perspective considering the interpretability research community place heavy emphasis on sparsity as a stand-in for interpretability itself. They show that some tasks, such as Indirect Object Identification on a synthetic dataset (TinyStories), can be handled by fewer internal connections once ablation is applied, as measured by Automatic Circuit Discovery. The results consistently indicate that forced ablation improves interpretability signals while modestly increasing perplexity. This more or less captures the central significance: it suggests interpretability need not strictly come from turning off large portions of neurons, but instead from targeted local gating. On the other hand, experiments are confined to TinyStories. The dataset is extremely small and might not reveal the full complexity of real-world language tasks, so the approach’s generalizability remains somewhat open. Although for a workshop submission, the paper can be seen as promising early results worthy of being shared with the community.

In terms of quality, I see no major flaws in data analysis or rigour. The clarity is fair, though a few parts could be expanded for a more thorough exposition of certain hyperparameters or complexities in global ablation. The originality is adequately demonstrated. The significance is moderate but could be higher if tested on bigger models or more challenging benchmarks (eg, unlearning as the authors point out). Still, it is a meaningful step toward bridging structured interpretability with normal transformer training, and more importantly, provide a perspective away from prioritizing sparsity at the cost of other metrics.

---

### Decision · Program_Chairs · 2025-03-05

Accept